# Hepatoprotection of a Standardized Extract of Cultured *Lentinula edodes* Mycelia against Liver Injury Induced by Ischemia-Reperfusion and Partial Hepatectomy

**DOI:** 10.3390/nu16020256

**Published:** 2024-01-14

**Authors:** Richi Nakatake, Tetsuya Okuyama, Morihiko Ishizaki, Hidesuke Yanagida, Hiroaki Kitade, Katsuhiko Yoshizawa, Mikio Nishizawa, Mitsugu Sekimoto

**Affiliations:** 1Department of Surgery, Kansai Medical University, Hirakata 573-1010, Osaka, Japan; okuyamat@hirakata.kmu.ac.jp (T.O.);; 2Department of Innovative Food Sciences, School of Food Sciences and Nutrition, Mukogawa Women’s University, 6-46 Ikebiraki-cho, Nishinomiya 663-8558, Hyogo, Japan; yoshizak@mukogawa-u.ac.jp; 3Department of Biomedical Sciences, College of Life Sciences, Ritsumeikan University, 1-1-1 Nojihigashi, Kusatsu 525-8577, Shiga, Japan; nishizaw@sk.ritsumei.ac.jp

**Keywords:** ECLM, hepatic ischemia-reperfusion injury, partial hepatectomy, inflammatory mediators, liver regeneration

## Abstract

A standardized extract of cultured *Lentinula edodes* mycelia (ECLM, AHCC^®^) has been shown to have beneficial effects on organ metabolism. ECLM has been indicated to have liver protective properties by suppressing inflammatory responses. The pathogenesis of hepatic ischemia-reperfusion injury is thought to involve the induction of inflammatory mediators. However, whether ECLM affects inflammatory mediators caused by warm hepatic ischemia-reperfusion injury and partial hepatectomy (HIRI+PH) has not been clarified. In this study, we evaluated the protective effects of ECLM against liver damage caused by HIRI+PH. Rats were fed a normal diet (HIRI+PH) or a normal diet with 2% ECLM (HIRI+PH and ECLM) for ten days, then the liver and duodenal ligament were clamped and subjected to 15 min of hepatic ischemia. After 70% hepatectomy, the inflow occlusion was released, and liver and blood samples were collected at 3, 6, and 24 h. The effect of ECLM on mortality induced by 30 min of ischemia and hepatectomy was evaluated. The results showed that ECLM attenuated pathological liver damage, including apoptosis, in the rats treated with HIRI+PH, and decreased serum aminotransferase activity; ECLM decreased mRNA levels of the inflammation-related genes inducible nitric oxide synthase and C-X-C motif chemokine ligand 1, and increased mRNA levels of interleukin 10, an anti-inflammatory cytokine; ECLM increased hepatocyte growth factor mRNA levels and Ki-67 labeled nuclei in the liver at 24 h; ECLM significantly reduced HIRI+PH-induced mortality. In conclusion, ECLM may prevent HIRI+PH-induced liver injury in part by suppressing various inflammatory responses and promoting liver regeneration.

## 1. Introduction

A number of studies have attempted to identify ways to prevent intraoperative hemorrhage during hepatectomy in human clinical settings. Inflow interception is one of the most effective methods of controlling bleeding during hepatic resection. First described by Pringle in 1908 [1], the Pringle maneuver (PM) is a simple and reproducible non-selective inflow occlusion technique for the portal triad [2] and has been widely used [3,4]. Warm hepatic ischemia-reperfusion injury (HIRI) is frequently encountered in hepatobiliary operations and is an important risk factor for postoperative complications in major hepatic resection [5]. Appropriate damage control of HIRI during liver resection is important to prevent complications such as liver failure and multiorgan damage. However, clinically effective treatments for HIRI are not available.

A standardized extract of cultured *Lentinula edodes* mycelia (ECLM, AHCC^®^), an extract from cultured *Lentinula edodes*, is produced by Amino Up Co., Ltd. (Sapporo, Japan) and is frequently used in patients with cancer in clinical practice [6]. ECLM contains a high amount of carbohydrates and a low amount of fiber [7]. The acetylated forms of α-1,4-glucan are thought to be the most active components and play a diverse role in cancer (i.e., pancreas [6], breast [8], and ovaries [9]), host protection during infections [10], or bacterial [11,12], and chronic diseases (i.e., diabetes) [13]. We previously isolated adenosine as another active component of ECLM and reported that ECLM and adenosine inhibit the activation of nuclear factor (NF)-κB (nuclear translocation and/or binding capacity to κB sites), resulting in the inhibition of nitric oxide (NO) production and inducible nitric oxide synthase (iNOS) induction in primary cultured rat hepatocytes stimulated by the inflammatory cytokine, an ex vivo model of liver injury [14,15]. ECLM also suppressed the production of inflammatory mediators in rat models of endotoxin-induced liver injury after partial hepatectomy (PH) [16] and small intestine ischemia-reperfusion injury [17]. These results suggest that ECLM may exert protective effects in warm HIRI. We hypothesized that pretreatment with ECLM would prevent inflammation caused by ischemia-reperfusion injury in the liver and be beneficial for subsequent liver regeneration. In the present study, the protective effects of ECLM were investigated using a warm hepatic ischemia-reperfusion injury and partial hepatectomy (HIRI+PH) model that mimics human clinical practice.

## 2. Materials and Methods

### 2.1. Materials and Animals

The ECLM was provided by Amino Up Co., Ltd. (Sapporo, Japan). In the manufacturing procedure, the mycelium source from a frozen working culture is incubated in the medium for more than approximately thirty-seven days, while being scaled up sequentially. The solution is centrifuged to remove insoluble cellular material. The various nutrient-rich supernatant is lyophilized to obtain the ECLM as a powder. For quality control of ECLM, a stable culture environment is maintained over a long time by strictly monitoring and controlling temperature, pH, brix, specific gravity, and microbial contamination. The manufacturing procedures follow the standard, Good Manufacturing Practice (GMP) for dietary supplements, prepared by the Ministry of Health, Labour and Welfare (MHLW) in Japan and certified by the Japan Health and Nutrition Food Association (JHNFA). International Organization for Standardization (ISO) 9001:2015 [18] for quality control and ISO 22000:2018 [19], which includes the Hazard Analysis Critical Control Point (HACCP) principles developed by the Codex Alimentarius Commission (CAC) and interactive communication, system management, prerequisite programs, management for safety management of the procedures are also certified. Together with the use of the frozen working culture, these will achieve the consistency of ECLM [7,20]. Male Sprague-Dawley rats (140–160 g; five weeks old) were purchased from Jackson Laboratory Japan, Inc. (Yokohama, Japan). The rats were kept at 22 °C on a 12 h light/dark cycle for ≥seven days to allow acclimatization before experiments were performed. The rats were allowed to take food and water. Animal care and experiments were conducted in accordance with the standards as outlined in the ARRIVE [21] and PREPARE [22] guidelines. The protocol in this study was approved by the Animal Care Committee of Kansai Medical University (Osaka, Japan) (approval no. 22-039 and no. 22-040).

### 2.2. Induction of HIRI+PH in Rats

After acclimatization, the rats were randomly divided into three groups (control, n = 3; HIRI+PH-only, n = 12; and HIRI+PH and ECLM, n = 14). The HIRI+PH-only rats received a normal diet and water for ten days, whereas the HIRI+PH and ECLM rats received a normal diet supplemented with 2% (*w*/*w*) ECLM and water for ten days. The administration route and the dosage of ECLM (a freely fed diet supplemented with 2% ECLM) were determined to be consistent with the previous in vivo studies [16,17]. The diet supplemented with 10% ECLM was refused by the rats in the previous in vivo studies. The rats were anesthetized with 3% isoflurane during PH with PM. The hepatoduodenal ligament was clamped using a vascular clip for a period of 15 min. After 70% hepatectomy, inflow occlusion was released by declamping. Liver and blood samples were obtained at 3, 6, and 24 h after declamping from the HIRI+PH-treated rats (each group, n = 4–6; Appendix A). As the control group, liver and blood samples were obtained from rats that were fed a normal diet without HIRI+PH treatment. In the survival experiment [23], the hepatoduodenal ligament clamp before PH was extended to 30 min. After 70% hepatectomy, inflow occlusion was released. To determine the survival rate of the rats, they were observed for seven days (HIRI+PH-only, n = 9; HIRI+PH and ECLM, n = 11; Appendix A). The NIH Office of Animal Care and Use [24] score and severity assessment were used to evaluate the animals after liver resection [25]. The rats were euthanized by treatment with high levels of isoflurane or by cervical dislocation when they became emaciated and morbid-looking with progressive liver failure, congestion, and multiple organ dysfunction.

### 2.3. Histopathological Analysis

The obtained liver specimens were fixed in phosphate buffered saline containing 4% paraformaldehyde and then embedded in paraffin. Sections were cut at 3–5 μm thickness using 3–5 pieces of liver per rat. To evaluate the pathological appearance of liver damage, they were stained with hematoxylin and eosin (H&E) and were graded on a score of 0–4 for sinusoidal congestion, cytoplasmic vacuolation of hepatocytes, and parenchymal necrosis, as described by Suzuki et al. [26]. The toxicologic pathologist (K.Y.) who was certified by the International Federation of Societies of Toxicologic Pathologists performed histopathological evaluations according to previous definition of diagnostic criteria and histopathological terminology. The result was expressed as the mean ± standard deviation (SD) of 3–6 rats per group. To detect apoptotic bodies in the hepatocyte nuclei, the in situ Apoptosis Detection Kit (Takara Bio Inc., Kusatsu, Shiga, Japan) based on terminal deoxynucleotidyl transferase-mediated deoxyuridine triphosphate-digoxigenin nick-end labeling (TUNEL) staining was used. To detect neutrophil infiltration in the liver tissues, the sections were stained with myeloperoxidase (MPO) using a rabbit polyclonal antibody against MPO (A0398; DAKO, Glostrup, Denmark) before counterstaining with hematoxylin. The number of TUNEL- and MPO-positive cells was counted in five fields of view of approximately 0.3 square millimeters per section by analysts blinded to the treatment arm, respectively. The numbers of positive cells per square millimeter in each field of view were calculated and the average was calculated for each rat. The results were expressed as the means ± SD of 3–4 rats per group. Liver regeneration was evaluated by staining the nuclei of proliferative hepatocytes with Ki-67 using a rabbit polyclonal antibody against Ki-67 (Leica Biosystems Newcastle Ltd., Newcastle upon Tyne, UK) before counterstaining with hematoxylin. The total nuclei and Ki-67-positive nuclei of hepatocytes were counted in five fields of view of 0.285 mm per section. The ratio of the number of stained nuclei to the total number of nuclei was calculated as a percentage of positive in each field of view and the average was calculated for each rat. The result was expressed as the mean ± SD of 4–6 rats per group.

### 2.4. Weight Change of the Remnant Liver

Liver regeneration after 24 h was estimated by the weight change rates of the remnant liver lobes: the weight of the resected liver lobes (equivalent to 70% of the total liver) was measured at the time of hepatectomy after PM and defined as W_A_. At euthanasia after HIRI+PH, the weight of the remnant liver lobes (equivalent to 30% of the total liver) was measured and defined as W_B_. The rate of change was calculated as [W_B_ − (W_A_/0.7) × 0.3]/W_B_.

### 2.5. Serum Biochemical Analysis

The activities of aspartate transaminase (AST) and alanine transaminase (ALT) in the serum were measured using the Transaminase C2-Test kit (code 431-30901, FUJIFILM Wako Pure Chemical Corp., Osaka, Japan). The serum was diluted with distilled water before measurement. The levels of nitrite and nitrate (NO_2_^−^ and NO_3_^−^, respectively, stable NO metabolites) in serum were measured using a Nitric Oxide Colorimetric Assay kit (Roche, Mannheim, Germany). Serum samples were diluted 1:2 in Reagent Diluent and filtered through a 10,000 molecular weight cut-off filter. The resulting serum samples were mixed with Nitrate Reductase enzymes and nitrate in the serum was reduced to nitrite. Total nitrite levels were measured as the amount of NO in the serum samples by the Griess method [27].

### 2.6. Reverse Transcriptase Polymerase Chain Reaction

Liver specimens were lysed with Kinematica Polytron PT 1300 D (Kinematica AG, Luzern, Switzerland) in the Sepasol I Super G (Nacalai tesque Inc., Kyoto, Japan), a guanidinium thiocyanate-phenol-chloroform mixture [28]. Then, total RNA was extracted from the lysate. Strand-specific reverse transcription polymerase chain reaction (RT-PCR) was conducted sequentially as follows; first, cDNA was prepared using an oligo (dT) primer for mRNAs from total RNA; touchdown quantitative PCR was performed using Rotor-Gene Q (Qiagen, Hilden, Germany) and the primer pairs listed in Appendix A to amplify cDNA for each gene. The mRNA levels were normalized to the levels of elongation factor-1α (EF) mRNA. The normalized value of each gene in HIRI+PH for 3 h was set as 1.0.

### 2.7. Statistical Analyses

Quantitative results were obtained from three to six rats per group per time-point. The mean values and SDs were calculated. For the comparison between the two groups, Student’s *t*-test was carried out. One-way analysis of variance (ANOVA) was carried out for multiple comparisons followed by Tukey–Kramer test to analyze differences. The Kaplan-Meier method was used to plot the cumulative survival curves of the HIRI+PH or HIRI+PH and ECLM groups. Meanwhile, the log-rank test was used to compare the difference in survival curves between the two groups. All statistical analyses and cumulative survival curve plots were performed using and JMP 17.2 statistics software (SAS Institute Inc., Carly, NC, USA). * *p* < 0.05 and ** *p* < 0.01 versus HIRI+PH alone at each time were considered significant.

## 3. Results

### 3.1. Pretreatment with ECLM Improved HIRI+PH-Triggered Liver Damage

Histological examination of the damage induced by HIRI+PH treatment was performed on rat liver specimens at 3 and 6 h after declamping (Appendix A). Figure 1A–D shows representative images of liver sections stained with H&E. Liver damage was characterized by infiltration of inflammatory cells and focal necrosis with hemorrhagic manifestations (Figure 1C). Ballooning degeneration in hepatocytes was also observed. ECLM reduced the incidence and extent of these pathological changes (Figure 1D). Damage was graded using the Suzuki score (Figure 1E), and ECLM significantly decreased liver damage at 6 h. Next, TUNEL staining of the liver sections showed that HIRI+PH increased hepatocyte apoptosis (Figure 1F), whereas ECLM markedly decreased apoptosis (Figure 1G,H). Furthermore, the HIRI+PH treatment markedly increased MPO-positive cells (Figure 1I). In contrast, ECLM significantly decreased MPO-positive cells (Figure 1J,K). A summary of these results is presented in Appendix A. Furthermore, HIRI+PH treatment increased serum AST and ALT activities, the liver injury markers. In contrast, ECLM administration significantly reduced serum AST and ALT activities (Figure 1L,M).

### 3.2. ECLM Affects Inflammation- and Apoptosis-Related Gene Expression in the HIRI+PH-Treated Rats

HIRI+PH treatment induced the mRNA expressions of several genes related to the inflammatory and apoptotic processes (Figure 2). Since these inflammation- and apoptosis-related genes are thought to directly affect liver injury, those expressions were analyzed at 3 and 6 h after the injury occurred. The mRNA expression levels of C-X-C motif chemokine ligand 1 (CXCL-1), an inflammatory chemokine [29], were increased in rat livers after HIRI+PH treatment, but ECLM significantly inhibited CXCL-1 mRNA expression (Figure 2A). In contrast, ECLM significantly enhanced the mRNA expression levels of interleukin-10 (IL-10), an anti-inflammatory cytokine [30,31], 3 h after declamping (Figure 2C). Moreover, the mRNA expression levels of myeloid cell leukemia1 (MCL1), an anti-apoptotic gene [32], were also significantly enhanced by ECLM (Figure 2D).

### 3.3. ECLM Improves Liver Regeneration in the HIRI+PH-Treated Rats

The regeneration of rat livers was investigated 24 h after HIRI+PH treatment. Evaluation of liver injury in the HIRI+PH and ECLM group showed no difference in the Suzuki score compared to the HIRI+PH group (Figure 3C); however, ECLM administration significantly increased the number of Ki-67 positive cells after HIRI+PH treatment (Figure 3F). A summary of the results is provided in Appendix A. The liver weights in the HIRI+PH and ECLM group were significantly higher than those in the HIRI+PH group (Figure 3G). To evaluate the liver regenerative effects of ECLM on the liver tissues, we assessed continuous changes in hepatocyte growth factor (HGF) mRNA expression levels up to 24 h after treatment. ECLM sustained a significantly higher level of HGF mRNA expression from 6 to 24 h after HIRI+PH treatment (Figure 3H).

### 3.4. ECLM Reduces NO Production and iNOS Induction and Elevates Endothelial Nitric Oxide Synthase (eNOS) Induction after HIRI+PH Treatment

Serum NO levels increased after HIRI+PH treatment. The HIRI+PH and ECLM group showed significant decreases (Figure 4A). In addition, HIRI+PH treatment increased iNOS mRNA expression levels in the livers; however, ECLM inhibited its expression after treatment (Figure 4B). In contrast, the mRNA expression levels of eNOS gene were enhanced by ECLM treatment at 6 h after HIRI+PH treatment (Figure 4C).

### 3.5. Effects of ECLM on HIRI-Induced Mortality

Finally, the protective effect of the ECLM under lethal conditions with prolonged ischemic duration (30 min) and PH was investigated (Appendix A). The survival rate in the HIRI+PH-only treatment group was only 0.593. However, ECLM administration significantly restored survival to 1.00 versus HIRI+PH-induced mortality (Figure 5).

## 4. Discussion

This study demonstrated the hepatoprotective effect of ECLM on HIRI, which is one of the primary contributors to morbidity and mortality after hepatic resection and liver transplantation [33]. After liver resection with warm HIRI, most patients are affected by invasion due to various factors, including a significant decrease in red blood cells and liver volume. This invasion induces an inflammatory response [34] and reactive oxygen species (ROS) and reactive nitrogen species are excessively produced in the remnant liver [35], resulting in fatal complications after hepatic resection, such as liver failure. Ischemia leads to adenosine triphosphate depletion and accumulation of toxic metabolites and reperfusion leads to ROS intermediates production, which causes oxidative stress [36]. The initial phase of HIRI is a Kupffer cell-mediated response where the release of many inflammatory cytokines is triggered by oxidative stress and infiltration of activated neutrophils into the liver parenchyma characterizes the second phase of HIRI [37,38]. ECLM attenuated the appearance of pathological changes (Figure 1A–E) in the livers and attenuated AST and ALT activities in the serum (Figure 1L,M) in the rat model of HIRI+PH. ECLM also decreased the mRNA expression of the inflammatory chemokine CXCL-1 (Figure 2A) and neutrophil infiltration (Figure 1K) but increased the mRNA expression levels of IL-10, the anti-inflammatory cytokine (Figure 2C). Furthermore, ECLM enhanced the expression of the anti-apoptotic factor MCL1 (Figure 2D) and suppressed hepatocyte apoptosis (Figure 1H). Taken together, these data strongly support the idea that ECLM exerts a protective effect on HIRI through inhibiting the processes of inflammation and apoptosis.

In this study, the potent effects of ECLM in promoting liver regeneration were also demonstrated. ECLM may have increased the liver weight after hepatectomy (Figure 3G) in part by increasing the proliferating hepatocytes that are positive for Ki-67 (Figure 3F) through maintaining increased HGF mRNA expression in the liver at 24 h after HIRI+PH treatment (Figure 3H). Liver failure after hepatic resection can be prevented by promoting liver regeneration [39]. Therefore, liver regeneration is an important prognostic factor in clinical settings including hepatic resection and liver transplantation. The mechanical basis of liver regeneration has been studied intensively [40], and several agents are still being studied for their liver regenerative effects. Adenosine administration promotes cell cycle progression in PH-induced rat liver regeneration [41], suggesting that adenosine-containing ECLM may be effective in stimulating liver regeneration. In rats, the process of liver regeneration initiates with the induction of diverse signaling molecules 3 h after hepatectomy, peaks at two to three days, and ends with the enlargement of the remnant liver lobes to the original liver size at five to seven days [42]. The liver regeneration-promoting effects of ECLM were evaluated at 24 h before reaching the peak of liver regeneration. These promoting effects did not appear to be related to the pathological appearance of liver injury; whether the promotion of liver regeneration by ECLM is related to the frequency of apoptosis or the infiltration of inflammatory cells is an important subject to be elucidated in future studies.

The role of NO in HIRI has been demonstrated to be exacerbated or attenuated by a variety of mechanisms [43,44]. NO, an unstable nitrogen-centered radical is catalyzed by NO synthase (NOS) in the reduction oxidation reaction in which molecular oxygen acts L-arginine; three types of NOS are known: eNOS, neuronal NOS (nNOS), and iNOS. eNOS-derived NO in hepatic sinusoidal endothelial cells is cytoprotective, whereas NO derived from iNOS is thought to act protectively or dramatically as an inflammatory mediator and contributes to pathological processes [45,46]. ECLM significantly increased eNOS mRNA expression (Figure 4C). The iNOS expression is clearly elevated in the warm HIRI [47] and ECLM significantly decreased iNOS mRNA induction and NO production (Figure 4A,B). In general, uncontrollable overproduction of NO by iNOS results in the liver injury caused by highly toxic compounds, including peroxynitrite and hydroxyl radicals, produced by the persistent reaction of large amounts of NO reacting with superoxide anions [48]. Regarding serum NO levels in the HIRI+PH and ECLM group, this decrease may contribute to the hepatoprotective effect. On the other hand, suppression of iNOS induction and NO production in hepatocyte cultures is considered to be an indicator of hepatoprotection [49]. In primary cultured rat hepatocytes, we previously reported the inhibitory effects of ECLM on the induction of iNOS gene and NO production, in part through inhibition of NF-κB activation [14,50]. However, no relationship was identified between iNOS downregulation and its impact on HIRI. Furthermore, no direct association between lower iNOS expression and longer survival in warm HIRI was identified [47]. The relationship between ECLM and iNOS in HIRI requires further investigations. In addition, direct evidence of whether downregulation of the expression of iNOS and the production of NO by ECLM contributes to hepatoprotection is currently uncertain and an important topic for future studies.

The administration of ECLM improved survival in the HIRI model under more severe conditions with an extended ischemia time of 30 min. ECLM may increase the survival rate through its anti-inflammatory and anti-apoptotic effects and by promoting liver regeneration. These results indicated that ECLM when administered preoperatively can be used in clinical practice as a therapeutic option to protect against liver injury after hepatic resection. The underlying mechanism of the hepatoprotective effects of ECLM, including the identification of its active ingredients, needs to be further investigated. Administration of ECLM, a functional food, is thought to affect the metabolism and gut microbiota of rats. Metabolomic analysis and microbial sequencing may provide a deeper understanding of the mechanisms of protective actions against liver injury induced by HIRI+PH treatment.

In clinical practice, impaired PM for more than 15 min in hepatic resection for hepatocellular carcinoma (HCC) increases the risk of HCC recurrence [51]. In rat models, HCC growth was also suggested to be caused by impaired reperfusion after long hepatic ischemia [52]. Therefore, HIRI may be involved in the recurrence of HCC. However, it has been suggested that ECLM may inhibit the recurrence of HCC [53,54]. It may be possible to apply the current investigation of the effect of ECLM on a warm HIRI+PH rat model to human clinical practice by observing the organ-protective effect for hepatectomy requiring frequent PM and the postoperative recurrence rate of HCC after ECLM oral intake. Further human clinical studies are necessary to clarify the effect of ECLM on PM and its relationship with the postoperative recurrence of HCC.

## 5. Conclusions

Oral ECLM administration attenuated the liver damage caused by PH with HIRI in a rat model. These effects may be associated with the suppression of inflammatory mediators, enhancement of anti-inflammatory cytokine and the anti-apoptotic factor expression, and promotion of liver regeneration, suggesting that ECLM may have therapeutic potential in the setting of hepatectomy requiring frequent ischemia-reperfusion to control bleeding.

## Figures and Tables

**Figure 1 nutrients-16-00256-f001:**
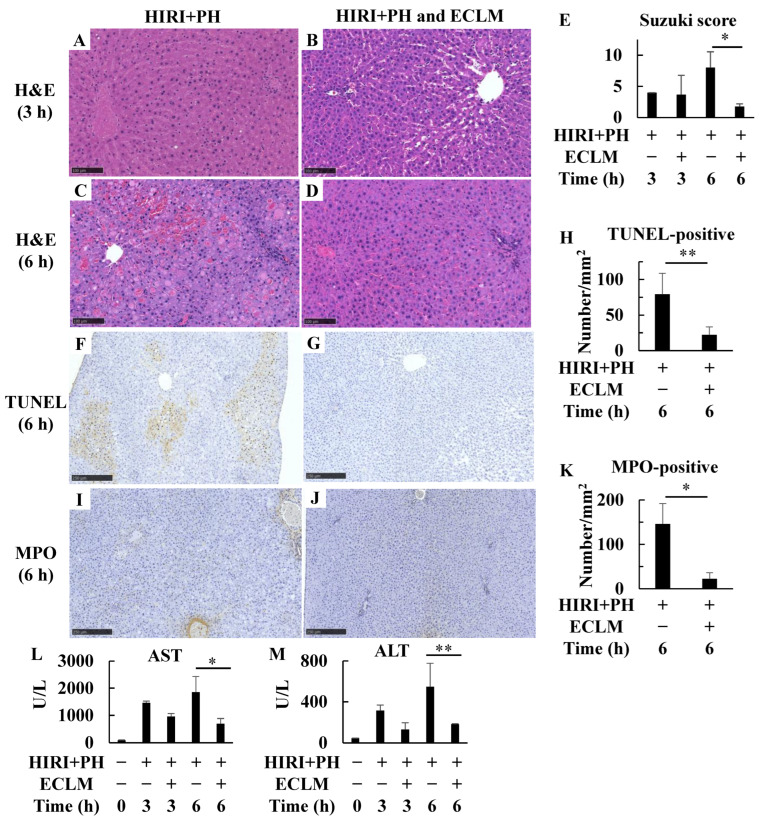
Effects of ECLM on pathological changes in the liver of HIRI+PH-treated rats. (**A**–**D**) Histologic appearance of the rat liver after HIRI+PH treatment. H&E-stained liver sections from rats fed the normal diet (HIRI+PH; (**A**,**C**)) and those fed with ECLM (HIRI+PH and ECLM; (**B**,**D**)) were obtained at 3 and 6 h after HIRI+PH treatment, respectively (magnification ×200; bar = 100 μm). In the HIRI+PH group, the areas of massive hemorrhage and focal necrosis with inflammatory cell infiltration are detected (**E**) Suzuki histological grading of the rat liver treated with HIRI+PH. Effects of ECLM on (**F**–**H**) apoptosis and (**I**–**K**) neutrophil infiltration in the liver. Liver sections obtained 6 h after HIRI+PH treatment from the HIRI+PH group (**F**,**I**) and the HIRI+PH and ECLM group (**G**,**J**) were stained (magnification ×100, bar = 250 μm). Positive cells show brown color in the nucleus or cytoplasm, respectively. The numbers of (**H**) TUNEL-positive and (**J**) MPO-positive cells per square millimeter were counted. Effects of ECLM on (**L**) AST and (**M**) ALT activities in the serum of HIRI+PH-treated rats. The values in the bar graphs represent the mean ± SD. * *p* < 0.05 and ** *p* < 0.01 versus HIRI+PH alone. ECLM, standardized extract of cultured *Lentinula edodes* mycelia; HIRI+PH, hepatic ischemia-reperfusion injury and partial hepatectomy; H&E, hematoxylin and eosin; TUNEL, terminal deoxynucleotidyl transferase-mediated deoxyuridine triphosphate-digoxigenin nick-end labeling; MPO, myeloperoxidase; AST, aspartate transaminase; ALT, alanine transaminase.

**Figure 2 nutrients-16-00256-f002:**
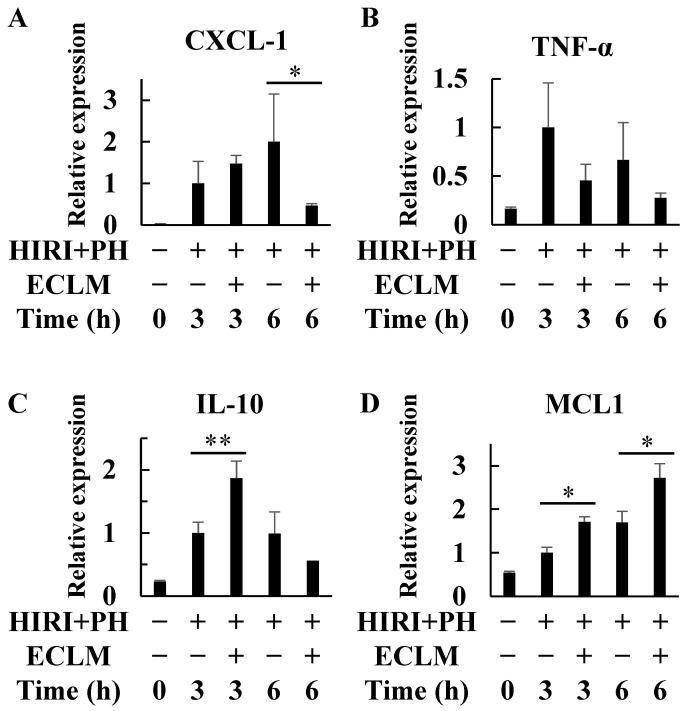
Effects of ECLM on the inflammation- and apoptosis-related gene expression in the livers of the HIRI+PH-treated rats. Total RNA was extracted from the liver at the indicated times after HIRI+PH treatment. Expression of (**A**) CXCL-1, (**B**) TNF-α, (**C**) IL-10, and (**D**) MCL1 mRNA was quantified by real-time RT-PCR. The values in the bar graphs represent the mean ± SD. * *p* < 0.05 and ** *p* < 0.01 versus HIRI+PH alone. ECLM, standardized extract of cultured *Lentinula edodes* mycelia; HIRI+PH, hepatic ischemia-reperfusion injury and partial hepatectomy; CXCL, C-X-C motif chemokine ligand; TNF tumor necrosis factor; IL interleukin; MCL, myeloid cell leukemia; RT-PCR, reverse transcriptase polymerase chain reaction.

**Figure 3 nutrients-16-00256-f003:**
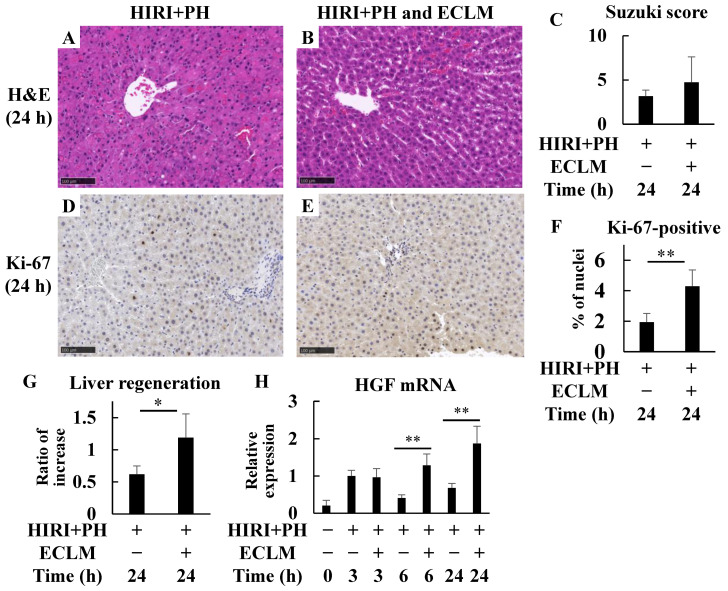
Effects of ECLM on hepatic regeneration in the HIRI+PH-treated rats. (**A**–**F**) Histologic appearance of the liver of rats treated with HIRI+PH after 24 h. The sections of the rats fed (**A**,**C**) the normal diet and (**B**,**D**) ECLM were stained with (**A**,**B**) H&E or (**C**,**D**) Ki-67 (magnification ×200; bar = 100 μm). (**C**) Suzuki histological grading of HIRI+PH. (**F**) Ki-67 labeling index calculated based on the number of positive nuclei of hepatocytes and the total number of nuclei. (**G**) Effects of ECLM on the liver regeneration rates 24 h after HIRI+PH treatment. The values in the bar graphs represent the mean ± SD. (**H**) Effects of ECLM on the mRNA expression levels of HGF in the livers. The values in the bar graphs represent the mean ± SD. * *p* < 0.05 and ** *p* < 0.01 versus HIRI+PH alone. ECLM, standardized extract of cultured *Lentinula edodes* mycelia; HIRI+PH, hepatic ischemia-reperfusion injury and partial hepatectomy; H&E, hematoxylin and eosin; HGF, hepatocyte growth factor.

**Figure 4 nutrients-16-00256-f004:**
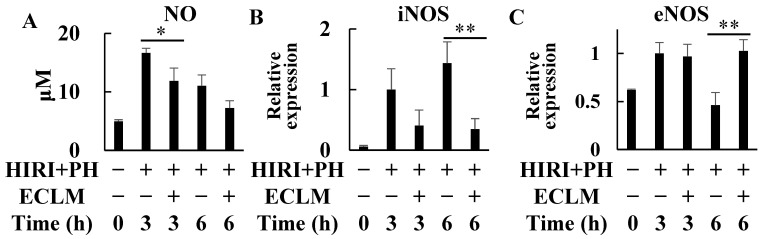
Effects of ECLM on NO levels in the serum and iNOS and eNOS mRNA expression levels in the rat livers after HIRI+PH treatment. (**A**) The total amount of nitrite and nitrate (NO_2_^−^ and NO_3_^−^, metabolites of NO) in the serum were measured. Total RNA from the liver of the HIRI+PH-treated rats was analyzed by real-time RT-PCR to quantify (**B**) iNOS and (**C**) eNOS mRNA expression levels. The values in the bar graphs represent the mean ± SD. * *p* < 0.05 and ** *p* < 0.01 versus HIRI+PH alone. ECLM, standardized extract of cultured *Lentinula edodes* mycelia; HIRI+PH, hepatic ischemia-reperfusion injury and partial hepatectomy; NO, nitric oxide; iNOS, inducible nitric oxide synthase; eNOS, endothelial nitric oxide synthase; RT-PCR, reverse transcriptase polymerase chain reaction.

**Figure 5 nutrients-16-00256-f005:**
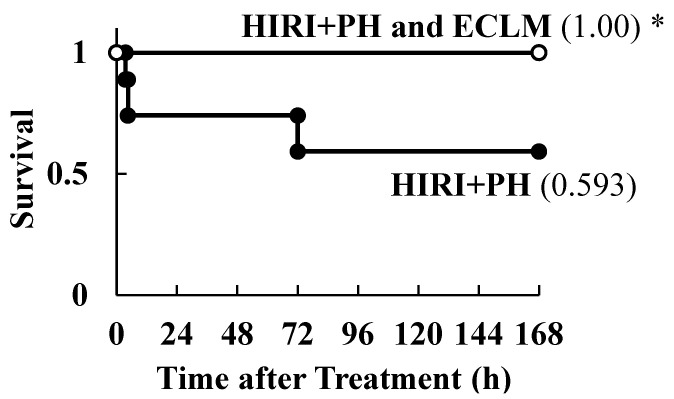
Effects of ECLM on rat survival in HIRI+PH treatment with a longer ischemic duration. The cumulative survival of the following treatment groups was plotted using Kaplan-Meier survival curves: HIRI+PH (filled circles; n = 9) and HIRI+PH and ECLM (open circles, n = 11). The values in parentheses represent survival at 168 h after HIRI+PH treatment. * *p* < 0.05 versus HIRI+PH alone. ECLM, standardized extract of cultured *Lentinula edodes* mycelia; HIRI+PH, hepatic ischemia-reperfusion injury and partial hepatectomy.

## Data Availability

The data presented in this study are available on request from the corresponding author.

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
