# Peer review of "Hepatoprotection of a Standardized Extract of Cultured Lentinula edodes Mycelia against Liver Injury Induced by Ischemia-Reperfusion and Partial Hepatectomy"

_nutrients, 2024, doi:10.3390/nu16020256_

Round 1

Reviewer 1 Report

Comments and Suggestions for Authors

In this paper, Nakatake and colleagues evaluated the protective effects of a dietary supplement derived from Lentinula edodes (ECLM) against hepatic ischemia/reperfusion followed by partial hepatectomy (HIRI+PH). The authors showed that the diet enriched with 2% ECLM attenuates liver damages, such as apoptosis, and reduced inflammation by decreasing the expression of pro-inflammatory molecules and promoting the expression of anti-inflammatory and anti-apoptotic molecules. Moreover, ECLM-enriched diet was able to reduce the mortality induced by a longer period of ischemia (30 min) before partial hepatectomy. This is a really interesting study, showing how the simple addition of a natural dietary supplement on a diet could have beneficial/protective effects against HIRI-induced liver damage. However, there some points the authors should address to strengthen the data before publication:

-  - The authors show the effects of ECLM in the diet on rats undergoing HIRI+PH versus rats receiving HIRI+PH without ECLM, but they are not showing the effects of this dietary supplement (ECLM only) on the liver of rats, particularly on TUNEL- and Ki-67 positive cells as well as ALT/AST levels and pro- and anti-inflammatory genes (CXCL-1, TNF-a, IL-10 and MCL1).

- - In figure 1, 2 and 4 they are showing effects of the ECML treatment 3- and 6-hours following HIRI+PH. They should add data of gene expression as well as ALT/AST levels, TUNEL assay and NO, iNOS and eNOS after 24 hours to show the improvement not only immediately after the damage but also in a more advanced phase, to rule out a possible delay of the liver damage as an effect of the diet enriched with ECLM. For example, IL-10 is higher in ECLM group after 3 hours but it is lower after 6 hours compared to no ECLM group, which shows stable IL-10 expression at these time points. Thus adding the 24h timepoint should help in understanding the outcome of the treatment.

-  - In figure 3C, Suzuki score is higher (although not significative) in ECLM group. Can the authors explain this result?

-  - In figure 3F they only show increased in Ki-67+ nuclei at 24h. They should perform TUNEL assay on these samples to show whether the cell death is reduced with ECLM treatment

-  - Figure 3H: please check the HGF mRNA expression levels in ECLM-only treated mice (no HIRI+PH)

-  - Figure 5: they should show the liver regeneration rates along with the survival curve to show the eventual improvement given by the ECLM-enriched diet

Other minor points:

-        Figure 3G and figure 5: please add the title to the axis.

-        Material and methods, page 3, line 107: “standard derivative”. Did they mean standard deviation?

-        It would help to add levels of mRNA and molecules analysed from a normal (untreated rat no surgery no ECLM) as a reference values.

Author Response

Reviewer1

First of all, we thank the reviewer for the invaluable comments and suggestions. We have addressed all the comments and revised the text in green. Please find our responses to the comments below.

In this paper, Nakatake and colleagues evaluated the protective effects of a dietary supplement derived from Lentinula edodes (ECLM) against hepatic ischemia/reperfusion followed by partial hepatectomy (HIRI+PH). The authors showed that the diet enriched with 2% ECLM attenuates liver damages, such as apoptosis, and reduced inflammation by decreasing the expression of pro-inflammatory molecules and promoting the expression of anti-inflammatory and anti-apoptotic molecules. Moreover, ECLM-enriched diet was able to reduce the mortality induced by a longer period of ischemia (30 min) before partial hepatectomy. This is a really interesting study, showing how the simple addition of a natural dietary supplement on a diet could have beneficial/protective effects against HIRI-induced liver damage. However, there some points the authors should address to strengthen the data before publication:

- The authors show the effects of ECLM in the diet on rats undergoing HIRI+PH versus rats receiving HIRI+PH without ECLM, but they are not showing the effects of this dietary supplement (ECLM only) on the liver of rats, particularly on TUNEL- and Ki-67 positive cells as well as ALT/AST levels and pro- and anti-inflammatory genes (CXCL-1, TNF-a, IL-10 and MCL1).

We previously reported the examination in the liver of normal rats which were administered with ECLM alone for inflammation-related genes (CXCL-1, TNF-a, and IL-10) (Nakatake R, et al. 2016); mRNA expression of these genes in the liver and gene products in serum were not detected in either the ECLM alone or the non-ECLM administration. On the other hand, the anti-apoptotic gene MCL1 is expected to be expressed to some extent under normal conditions. We analyzed the mRNA expression using Liver samples that had been dissected before perfusion and in which no damage had been induced and showed little effect of ECLM. In addition, normal mice supplemented with ECLM showed no significant change in plasma AST or ALT levels compared to the control group (Wakame K, et al. 2016). Furthermore, in safety assessment, no abnormalities such as hypertrophy, carcinogenesis, or liver damage have been observed in the liver after long-term administration of ECLM to normal rats or mice (Walshe, T. and Nishioka, H. 2016). Therefore, in this study, the pathological analyses performed in the livers of rats treated with ECLM alone. A description of the safety assessment was added to the Methods. These unperformed experiments may provide important insights into the safety of ECLM and its hepatoprotective mechanism of action. However, it was expected to be difficult to obtain results to respond to peer review within 10 days.

Nakatake, R.; Tanaka, Y.; Ueyama, Y.; Miki, H.; Ishizaki, M.; Matsui, K.; Kaibori, M.; Okumura, T.; Nishizawa, M.; Kon, M. Protective effects of active hexose correlated compound in a rat model of liver injury after hepatectomy. Funct Foods Health D. 2016, 6, 702-717.

Wakame K, Nakata A, Sato K, Miura T, Kulkarni AD, et al. (2016) DNA microarray analysis of gene expression changes in ICR mouse liver following treatment with active hexose correlated compound Integr Mol Med 3: DOI: 10.15761/IMM.1000235.

Walshe, T.; Nishioka, H. Safety Assessment. In Clinician's Guide to AHCC: Evidence-Based Nutritional Immunotherapy; Kulkarni, A.D., Calder, P., Ito, T., Eds.; International Congress on Nutrition and Integrative Medicine: Sapporo, Japan, 2016; pp. 34–38.

In figure 1, 2 and 4 they are showing effects of the ECML treatment 3- and 6-hours following HIRI+PH. They should add data of gene expression as well as ALT/AST levels, TUNEL assay and NO, iNOS and eNOS after 24 hours to show the improvement not only immediately after the damage but also in a more advanced phase, to rule out a possible delay of the liver damage as an effect of the diet enriched with ECLM. For example, IL-10 is higher in ECLM group after 3 hours but it is lower after 6 hours compared to no ECLM group, which shows stable IL-10 expression at these time points. Thus adding the 24h timepoint should help in understanding the outcome of the treatment.

Serum ALT/AST and gene expression of IL-10, MCL1, iNOS, and eNOS were examined in the samples at 24 h after HIRI+PH treatment; the effect of ECLM was not significant.

-  In figure 3C, Suzuki score is higher (although not significative) in ECLM group. Can the authors explain this result?

In Figure 3C, the ECLM group appeared to have a slightly higher Suzuki score than the control group, but also had a larger standard deviation. This indicates that individual differences were greater.

- In figure 3F they only show increased in Ki-67+ nuclei at 24h. They should perform TUNEL assay on these samples to show whether the cell death is reduced with ECLM treatment

We first evaluated using the Suzuki score, which includes an assessment of necrosis, at 3, 6, and 24 hours and found a significant difference at 6 h. Therefore, TUNEL staining was performed at 6 h samples and showed a significant decrease in positive cells by ECLM. We assumed that there would be no significant difference by TUNEL staining at 24 h. However, we agree that examining the relationship between ameliorating liver injury and promoting liver regeneration would provide a new dimension to the research. We added the description about the importance and these future subject in the discussion.

Lines: 327-330

- Figure 3H: please check the HGF mRNA expression levels in ECLM-only treated mice (no HIRI+PH)

In rats fed normal diet or ECLM, the mRNA expression was analyzed using liver samples that were dissected by partial hepatectomy immediately after ischemia and before perfusion. No damage was considered to be induced in the dissected liver lobes. Little effect of ECLM was observed (Figure 3H, Time: 0 h). The liver lobe used for these samples differs from the remnant liver lobe used for samples at other time points. To avoid misunderstandings, we made the following modifications to the graphs; the liver sample values (Time: 0 h) were excluded from Figure 3H: the values for the control group with no HIRI+PH and no ECLM were corrected to the Time: 0 h values (previously labeled as 'Naive' in the old graph).

Figure 5: they should show the liver regeneration rates along with the survival curve to show the eventual improvement given by the ECLM-enriched diet

We agree that it is important to show the rate of liver regeneration to evaluate the improvement of liver injury. However, in the survival experiments, because approximately half of the rats died due to the lethal conditions induced by the prolonged ischemic time, no significant difference in liver regeneration rate with ECLM was found.

Other minor points:

Figure 3G and figure 5: please add the title to the axis.

We added "Ratio of increase" and "Survival" to the vertical axes of Figure 3G and Figure 5, respectively.

Material and methods, page 3, line 107: “standard derivative”. Did they mean standard deviation?

As the reviewer pointed out, "standard deviation" is correct. We corrected the text in the Methods.

Line: 126

-It would help to add levels of mRNA and molecules analysed from a normal (untreated rat no surgery no ECLM) as a reference values.

Samples from normal rats had already been analyzed. In the graphs representing the analysis of gene expression and serum molecules, the values of HIRI+PH-, ECLM-, and 0 h correspond to that for normal rats. The description of sample collection from normal rats in Methods was revised.

Lines: 106-108

Reviewer 2 Report

Comments and Suggestions for Authors

The article proposed by Nakatake et al. focuses on the beneficial effect of a standardized extract of cultured Lentinula edodes mycelia (ECLM) against hepatic ischemia-reperfusion injury and partial hepatectomy.  The authors selected an in vivo rat model through which evaluate the protective effect of ECLM, including it on the diet (supplemented with 2% w/w of extract) of the corresponding treatment group. Effectively, the aforementioned model results in a perfect strategic choice to mimic the eventual damage due to organ reperfusion during hepatectomy.

The variety of data shown look sufficiently acceptable and abundantly corroborate the hypothesis according to which ECLM exerts protective effects towards post-reperfusion hepatic performance. In this regard, ECLM biological properties seem to be supported by many different investigations including histopathological assay or serum and gene expression analysis.

I have just a question about histopathological results shown in Figure 1: did the authors perform TUNEL and MPO staining also after 24h since partial hepatectomy? If so, did you find any change?

Oppositely, I perfectly agree with the time point selected to perform both inflammatory and liver regenerative panel.

I appreciated the different biological perspectives evaluated by authors to scientifically explain the role of the extract in liver-ischemia reperfusion damage.

Overall, I personally consider the manuscript a really well-structured paper, rich in interesting scientific evidences absolutely expendable for future applications in clinical practice.

Author Response

Thank you very much for providing important insights. We are grateful for the time and energy you expended on our behalf.

You will find our responses to each of your points and suggestions.

I have just a question about histopathological results shown in Figure 1: did the authors perform TUNEL and MPO staining also after 24h since partial hepatectomy? If so, did you find any change?

We first evaluated HIRI+PH-induced liver injury at 3, 6, and 24 h using Suzuki score, which also includes an assessment of hepatocyte necrosis and found that ECLM showed significant suppression at 6 h. Therefore, TUNEL staining as well as MPO staining was performed only at 6 h and both showed a significant decrease in positive cells by ECLM. It was assumed that there would be no significant difference in TUNEL and MPO staining at 24 h. However, we agree that examining the relationship between ameliorating liver injury and promoting liver regeneration at 24 h would be important. We added the description about the importance and these future subject in the Discussion.

Lines: 327-330

Reviewer 3 Report

Comments and Suggestions for Authors

This article, titled " Effect of A Standardized Extract of Cultured Lentinula edodes Mycelia on Liver Ischemia-Reperfusion" establishes a model of liver ischemia-reperfusion in rats and demonstrates improvement through the administration of Standardized Extract of Cultured Lentinula edodes Mycelia, which holds some clinical significance. However, there are some concerns that need to be addressed:

  1. The introduction lacks sufficient detail about Lentinula edodes Mycelia, such as its extraction source and primary components.
  2. What do you mean of “Standardized Extract”? How to define “Standardized”?
  3. Between lines 48-52, which components in the extract and adenosine exhibit a protective effect on the liver?
  4. What type of rats serve as experimental animals, and how many rats are involved?
  5. How are the rats euthanized?
  6. The term "naïve group" needs clarification. The article states that naïve and HIRI+PH-only rats were given water and a normal diet for 10 days, but the difference between them is unclear. Additionally, the results of the naïve group did not appear in later experiments.
  7. The authors' study lays the groundwork for understanding the protective effect of Lentinula edodes Mycelia in liver ischemia-reperfusion. I recommend additional experiments, such as metabolomic or microbial sequencing, to delve into the mechanism of its protective action if the authors have the resources for follow-up.
  8. Does liver ischemia-reperfusion have specific side effects? Are there any relevant reports on this matter?

 Author Response

First of all, we thank the reviewer for the invaluable comments and suggestions. We have addressed all the comments and revised the text in green. Please find our responses to the comments below.

This article, titled " Effect of A Standardized Extract of Cultured Lentinula edodes Mycelia on Liver Ischemia-Reperfusion" establishes a model of liver ischemia-reperfusion in rats and demonstrates improvement through the administration of Standardized Extract of Cultured Lentinula edodes Mycelia, which holds some clinical significance. However, there are some concerns that need to be addressed:

  1. The introduction lacks sufficient detail about Lentinula edodes Mycelia, such as its extraction source and primary components.

As described in the literature below, the manufacturing process and composition of ECLM were added to the Materials and Methods and the Introduction, respectively.

Lines: 71-75

Lines: 51-55

Fujii, N.; Kudo, S. Manufacturing Process. In Clinician's Guide to AHCC: Evidence-Based Nutritional Immunotherapy; Kulkarni, A.D., Calder, P., Ito, T., Eds.; International Congress on Nutrition and Integrative Medicine: Sapporo, Japan, 2016; pp. 21–23.

Sato, K.; Kashimoto, M. Composition. In Clinician's Guide to AHCC: Evidence-Based Nutritional Immunotherapy; Kulkarni, A.D., Calder, P., Ito, T., Eds.; International Congress on Nutrition and Integrative Medicine: Sapporo, Japan, 2016; pp. 24–33.

  1. What do you mean of “Standardized Extract”? How to define “Standardized”?

As described in the literature below, the detail of quality control on the manufacturing process of ECLM was added to Materials and Methods.

Lines 75-86

Fujii, N.; Kudo, S. Manufacturing Process. In Clinician's Guide to AHCC: Evidence-Based Nutritional Immunotherapy; Kulkarni, A.D., Calder, P., Ito, T., Eds.; International Congress on Nutrition and Integrative Medicine: Sapporo, Japan, 2016; pp. 21–23.

  1. Between lines 48-52, which components in the extract and adenosine exhibit a protective effect on the liver?

Adenosine is one component isolated from ECLM. We reported that adenosine inhibits the induction of inflammatory mediator production using primary cultured rat hepatocytes stimulated with IL-1β, a model of in vitro liver injury. The text was revised because the wording could be misleading.

Lines: 55-60

  1. What type of rats serve as experimental animals, and how many rats are involved?

After acclimation, 6-week-old male Sprague-Dawley rats were randomly assigned: 3 to the naive group, 21 to the HIRI+PH only group (9 of which were in survival experiments), and 25 to the HIRI+PH and ECLM group (11 of which were in survival experiments). The number of rats assigned was added to the Materials and Methods.

Lines: 95-96

  1. How are the rats euthanized?

Rats were euthanized by treatment with high levels of isoflurane or by cervical dislocation. These were added to Methods.

Lines: 113-114

  1. The term "naïve group" needs clarification. The article states that naïve and HIRI+PH-only rats were given water and a normal diet for 10 days, but the difference between them is unclear. Additionally, the results of the naïve group did not appear in later experiments.

The term "naïve group" had referred to rats fed a normal diet without HIRI+PH treatment and corresponded to the values labeled HIRI+PH, -; ECLM, -; Time (h), 0 in each bar graph in Figures 1, 2, and 4. To avoid any misleading expressions, we newly defined rats fed a normal diet without HIRI+PH treatment as the "control group". The description in Methods has been corrected.

Lines: 106-108

  1. The authors' study lays the groundwork for understanding the protective effect of Lentinula edodes Mycelia in liver ischemia-reperfusion. I recommend additional experiments, such as metabolomic or microbial sequencing, to delve into the mechanism of its protective action if the authors have the resources for follow-up.

 -We have no resources for follow-up. However, as the reviewer pointed out, these experiments would bring a new stage to delve into the protective mechanism. We mentioned these promising experiments in the Discussion.

Lines: 362-365

Administration of ECLM, a functional food, is thought to affect the metabolism and gut microbiota of rats. Metabolomic analysis and microbial sequencing may provide a deeper understanding of the mechanisms of protective actions against liver injury induced by HIRI+PH treatment. 

  1. Does liver ischemia-reperfusion have specific side effects? Are there any relevant reports on this matter?

In the previous literature, there are no specific items other than those investigated for this study. However, the concept and evidence for survival prolongation and organ-protective effects of reducing NO are controversial and difficult to assess. These explanations were described in the text, Lines: 331-354.

Round 2

Reviewer 1 Report

Comments and Suggestions for Authors

The authors addressed all my points. I repeat myself: This is a really interesting study, showing how the simple addition of a natural dietary supplement on a diet could have beneficial/protective effects against HIRI-induced liver damage and therefore a significative impact in clinic.